# Improving risk assessments in conservation ecology

Kotaro Ono [1,2], Øystein Langangen [2] & Nils Chr. Stenseth[1,2]

Conservation efforts and management decisions on the living environment of our planet often rely on the results from statistical models. Yet, these models are imperfect and quantification of risk associated with the estimate of management-relevant quantities becomes crucial in providing robust advice. Here we demonstrate that estimates of risk themselves could be substantially biased but by combining data fitting with an extensive simulation–estimation procedure, one can back-calculate the correct values. We apply the method to 627 time series of population abundance across four taxa using the Gompertz state-space model as an example. We find that the risk of large bias in population status estimate increases with the species' growth rate, population variability, weaker density dependence, and shorter time series, across taxa. We urge scientists dealing with conservation and management to adopt a similar approach to ensure a more accurate estimate of risk measures and contribute towards a precautionary approach to management.

[1] Centre for Coastal Research (CCR), University of Agder, P.O. Box 422N-4604 Kristiansand, Norway. [2] Centre for Ecological and Evolutionary Synthesis (CEES), Department of Biosciences, University of Oslo, P.O. Box 1066 BlindernN-0316 Oslo, Norway. Correspondence and requests for materials should be addressed to K.O. (email: kotaro.ono@ibv.uio.no) or to Ø.L. (email: oystein.langangen@ibv.uio.no) or to N.C.S. (email: n.c.stenseth@mn.uio.no)

**O**ne of the main challenges and goals in conservation ecology is to sustainably manage the integrity of the ecosystem. A healthy ecosystem provides numerous services to the human population including food, climate regulation, disease regulation, nutrient cycling, and cultural experience[1]. In order to manage an ecosystem sustainably with its wildlife and surrounding environment, sound management decisions based on the knowledge of the structure and dynamics of the ecosystem are required[2]. Statistical models have often served as a workhorse behind many conservation or management decisions by providing important estimates of population status. These include examples of IUCN red list definition[3], endangered species listing in the US or Canada[4], fisheries catch limits estimation[5], and more. However, models are mere simplifications of the complex dynamics of the natural system[6]. Therefore, appropriate consideration of uncertainty is crucial to avoid fallacious interpretation of ecosystem functioning[7,8]. Sources of uncertainty represent, for example, our lack of knowledge on the mechanisms that govern the population dynamics or errors associated with measurements of the population. In this regard, it has become increasing popular and important to quantify risks (used in this study as the probability that something harmful to the society might happen e.g., risk of overexploitation, risk of population extinction) in population and conservation ecology[9,10] and many institutions as well as regulations now require a precautionary approach to conservation and management[11].

However, estimates of risk may themselves be biased in statistical models—even in ones that are carefully configured[12]. This can have important consequences if these estimates are directly used to provide management advice. Much work has gone into evaluating model performance and parameter estimation bias[13,14] or evaluating risks of management strategies[8,15] but little has been directed to develop approaches that can correct for the inherent bias in risk estimates and calculate the correct level of risk associated with the study system. Indeed, all statistical models are, to some degree, subject to estimation bias and all model parameters (and derived quantities) have a probability of being estimated as a different set of values, depending on the amount of noise in the data, time-series length, and the statistical properties of the model.

Here we focus on a risk measure with management relevance: the probability that the estimate of final-year population depletion is at least 50% biased i.e., an absolute relative error rate of 0.5. Population depletion estimate—estimate of population size at the end of the time series compared to the population size at the start of the time series (or another reference point)—is often used to assess population status in fisheries and set catch limit accordingly i.e., if this ratio is below a given limit, no harvest is allowed, but when this ratio is high, a higher catch is allowed. The so-called 40–10 rule used in the US West coast by the Pacific Fishery Management Council is an example. We refer to this risk measure as risk of biased population status estimate. Through an extensive simulation–estimation procedure (>5 million datasets were generated, Fig. 1 and Methods), we quantify the risk of biased population status estimate for 627 time series of population abundance from the Global Population Dynamics Database (GPDD), covering four taxonomic groups[16].

Briefly, our approach consists in creating an extensive set of realistic population dynamics scenarios, appropriate for wildlife populations, from which we simulate data then perform data fitting. Next, we use the results from this simulation–estimation routine to back-calculate the probability that certain parameter combinations can be estimated as a different set of values (that we call the "true" parameter range). This probability depends on the parameter combination itself but also on the time-series length. We demonstrate our approach by using the Gompertz state-space

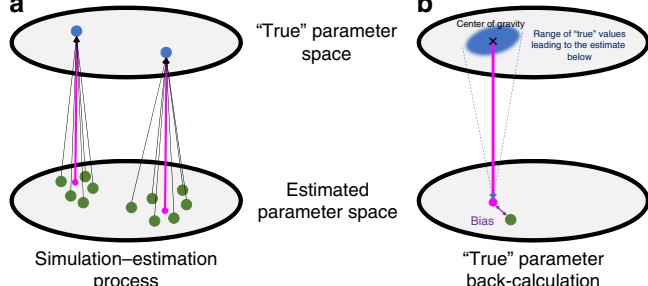

**Fig. 1** Schematic illustration of the simulation–estimation and parameter back-calculation. Schematic of **a** the simulation–estimation procedure and **b** back-calculation of the "true" parameter range that can produce the estimated values. Points in blue are the "true" parameter values used in the simulation (summarized here in a theoretical 2D surface). Points in green are the estimated parameter values (maximum likelihood estimate) when model are fitted to data generated from the corresponding simulation scenario (i.e., the blue points where the black arrows point towards in **a**). The magenta line with the magenta point in **a** shows the location of the unbiased parameters estimates. Any difference between the magenta and green points suggest parameter bias. In panel **b**, the shaded blue area in the "true" parameter space shows the range of estimated "true" parameter values that could have led to the green point in the estimated parameter space based on the simulation–estimation results from **a**. Additionally, in panel **b**, X points at the center of gravity of the "true" parameter space, the magenta point is the corresponding projection in the estimated parameter space, and the purple arrow shows the average bias

model (GSSM) as an example and show that the risk of biased population status estimate could be substantial for data-limited species with high growth potential, high population variability, and with weak density dependence. Gompertz model have been extensively used in ecology to study the dynamics of many different taxa[17–19] due to its well-known statistical properties, ability to capture important population processes such as density dependence (i.e., how important population processes change over time in relation to the population density) and different sources of uncertainty, as well as the fact that it does not require extensive quantities of data. Our approach is, however, applicable to many other ecological models, such as the Ricker model, theta-logistic model, and more complex models.

## Results

**Factors influencing the accuracy of risk estimates.** Using the example of the GSSM, we find that the underlying risk of biased population status estimate varies depending on the species life history and length of the time series (Supplementary Figs. 1–3). In general, species with high growth potential, high population variability, and weak density dependence tend to have a higher risk level than species with low growth potential, low variability, and high density dependence (Fig. 2a–e). High population variability adds noise to the data, which complicates parameter estimation[20]. Similarly, shorter time series complicate estimation as limited information is available to separate the signal from noise[17]. Species with weaker density dependence have increasing bias in the estimates of growth parameters (Supplementary Fig. 4), which leads to more bias in the estimate of final-year depletion level, and more risk (Fig. 2a–e).

On the other hand, estimates of density dependence are more biased for species with strong density dependence[21] (Supplementary Fig. 5). Finally, higher growth potential increases the risk of biased population status estimate because of a scaling issue: the GSSM population abundance is log transformed for

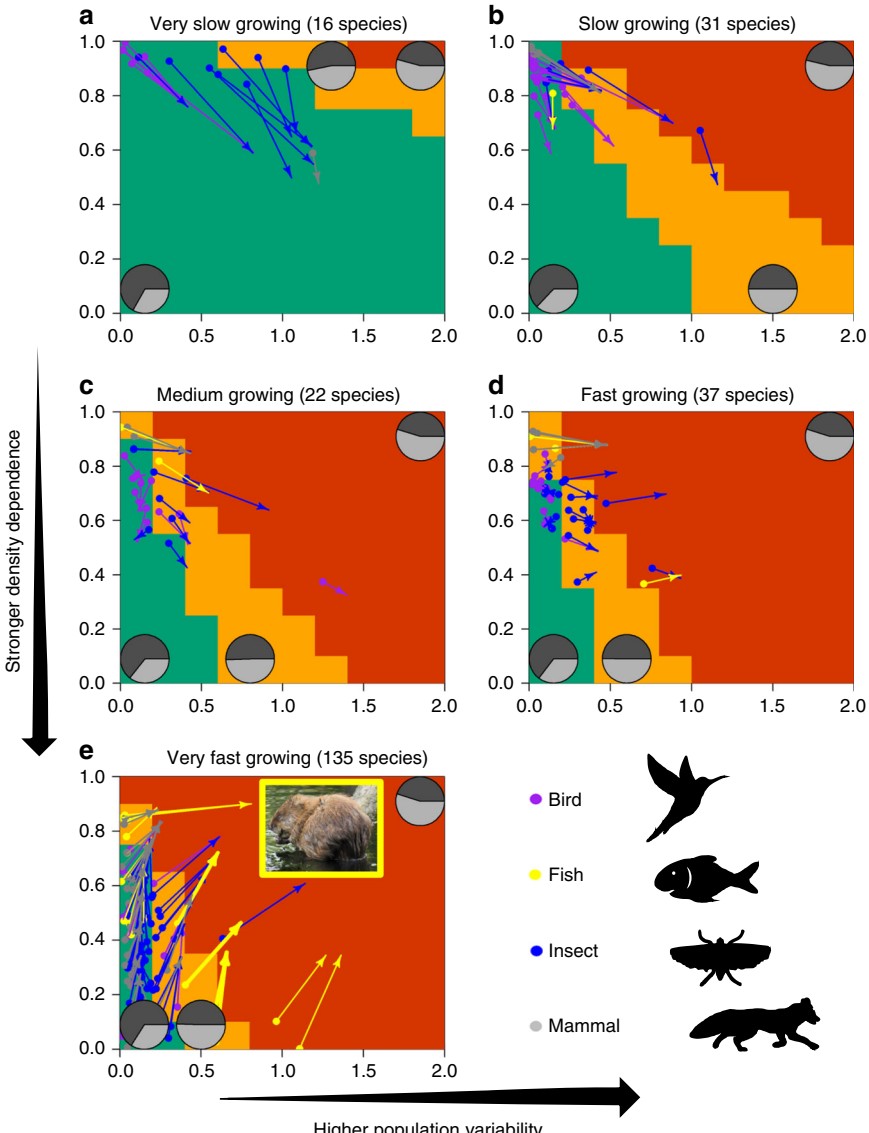

**Fig. 2** Risk of biased population status estimate. Risk maps of biased population status estimate across four taxonomic groups (birds, fish, insects, and mammals) with a time series length ranging from 21 to 30 years. Risks are quantified as having more than 60% (orange-red), 30–60% (orange), and below 30% (bluish-green) chance of estimating the final-year depletion level (how much the population has changed compared to the start of the time series or another reference point) with at least 50% bias in both directions. Risk are quantified based on the results from extensive (135,000 scenarios) simulation–estimation study. The risk map is summarized in 2D based on total (both process and observation) population variability measured in terms of coefficient of variation and population density dependence. The plot is restricted to the parameter range used in the simulation studies. The panels are organized by the species intrinsic rate of growth from very slow-growing (0–0.3) (**a**), slow-growing (0.3–0.7) (**b**), medium-growing (0.7–1.1) (**c**), fast-growing (1.1–1.5) (**d**), to very fast-growing (>1.5)) (**e**) species. The filled dots are the estimated parameter values for the four taxonomic groups (birds (purple), fish (gray), insects (blue), and mammals (yellow)) and the arrows show the most plausible ("true") parameter values that could have generated such estimates. The pie chart within each panel identifies the sign of estimation bias for each risk category i.e., false-positive (thinking the population is more than 50% healthier than it really is) in dark gray and false-negative (thinking the population is more than 50% in poorer condition that it actually is) in light gray. The thicker yellow arrows illustrate the example of muskrat (photo credit: Wikimedia Commons) from different regions of Canada

computational efficiency and accuracy but when back-transformed discrepancies in the estimates of bias are created e.g., underestimating by 10% a population size of 2 (in log scale) leads to a negative bias of 18% in original scale, whereas underestimating by 10% a population size of 10 (in log scale) leads to a negative bias of 63% in original scale.

**Directionality of bias**. There were more false-positives in risk estimates (i.e., thinking that population is healthier than it

actually is by at least 50%) for species in the lower risk category (i.e., species with lower growth rate, lower population variability, and stronger density dependence) but the proportion of false-positives decreases with the species risk category (Fig. 2a–e, Supplementary Figs. 2–3). This suggests that estimates of population depletion for species categorized in the lower risk zone have a low chance (<30%) of being very biased (>50% bias) but when they are, they have close to 75% chance of overestimating population status. An over-estimation of the population status may lead to an elevated

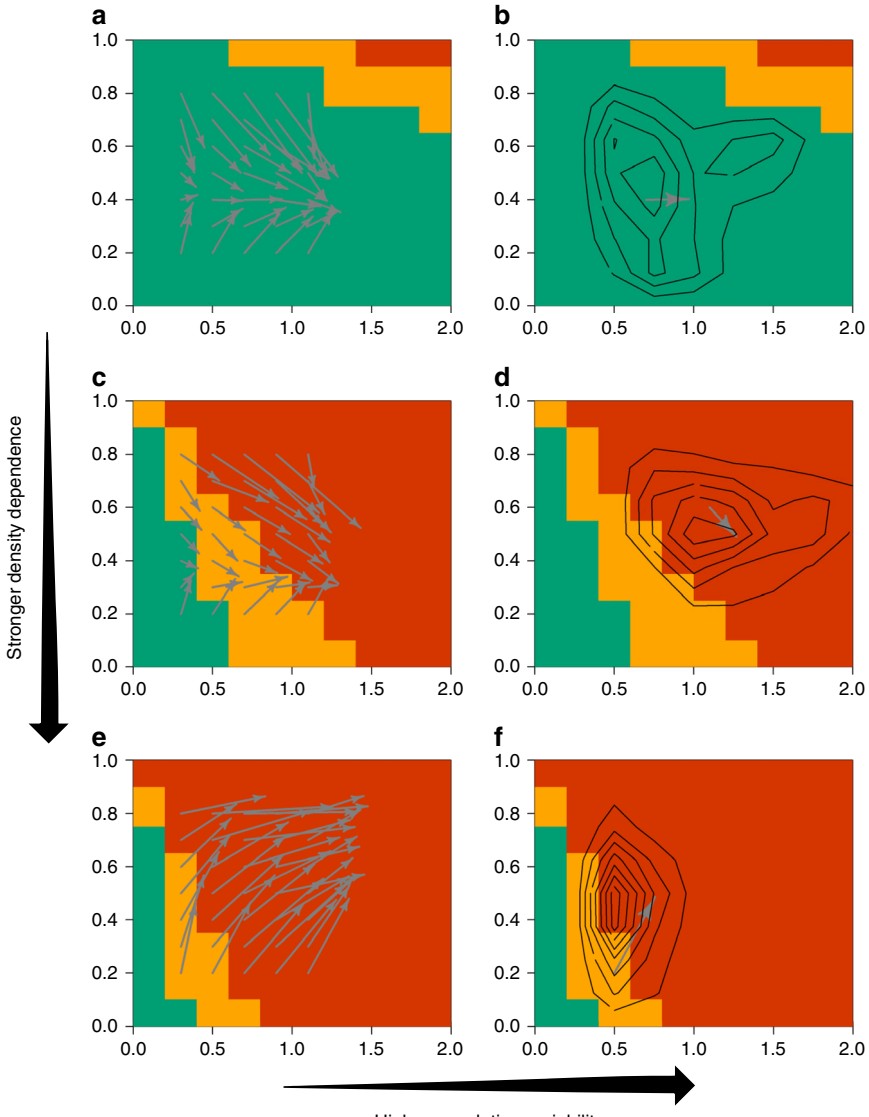

**Fig. 3** Directionality and strength of bias in risk estimates. The directionality and strength of bias in risk (of large bias in population status) estimates for different combination of population variability and density dependence for **a** very slow-growing species, **c** medium-growing species, and **e** very fast-growing species with a time series length ranging from 21 to 30 years. The longer the arrow, the stronger the bias. In this illustration, the base of the arrow shows the point estimates based on the GSSM fit to data and the arrow points towards the center of gravity of the "true" parameter range that likely generated such values based on simulation. The underlying color-coded risk map (orange-red >60%, orange between 30–60%, and bluish-green <30% chance of estimating the final-year depletion level with at least 50% bias) is represented based on the total (both process and observation) population variability measured in terms of coefficient of variation and population density dependence. The plot is restricted to the parameter range used in the simulation studies. **b**, **d**, **f** Contour plots of the back-calculated "true" parameter range for a few examples of very slow-, medium-, and very fast-growing species. Contour lines represent the probability density with higher density at the center of the contour

risk of population reduction (if population depletion estimate is used for setting species harvest limit e.g., the so-called 40–10 rule in fisheries) with negative consequences for conservation. Conversely, an underestimation of population status may lead to underestimation of the harvest potential of the population with negative socio-economic consequences.

We also find that the estimated risk of biased population status is often biased and the direction of bias is mostly influenced by the species growth potential, when using GSSM. The "true" risk level of a species with high growth rate tends to be larger than the one estimated from data (arrows pointing more toward the orange-red zone), while the risk for a species with lower growth rate tends to be lower (arrows pointing more toward

the bluish-green zone), independently of the taxa (Fig. 3a, c, e). As an example, three local populations of muskrat (*Ondatra zibethicus*)—a semi-aquatic rodent native from North America that is important to wetland ecology (due to its large grazing capability) and to the local economy (thanks to its fur)[22]—in Canada are all estimated to be at medium risk but the simulation–estimation exercise indicates that species with such parameter estimates and time series length should have a much higher risk on average (Fig. 2e). This suggests that the estimates of final-year population depletion level for these three rodents' populations are more likely to be highly biased (at least 50%) and would therefore require additional precaution if managing the populations based on such estimates.

**Back-calculating the "true" parameter range**. The "true" parameter values that likely generated the observed parameter estimates are quite uncertain and encompass varying degree of risk level (Fig. 3b, d, f). For example, species with similar parameter estimates to the three rodents mentioned above (i.e. the bottom left region of Fig. 3f) have roughly a 50–50 chance of falling either in the medium- or high- risk categories, whereas the most likely values fall in the high-risk zone. Furthermore, certain parameter combinations can even originate from a multimodal surface (Fig. 3b). All of the above observations suggest that rigorous risk evaluation needs to consider the full distribution of back-calculated "true" parameter values.

Concomitantly, future studies should also consider other sources of uncertainty such as parameter estimation uncertainty (see Supplementary Fig. 8 for illustrative example) or model types uncertainty (e.g., using a theta-logistic model in addition to GSSM) when back-calculating risk values. One can do so by taking a Bayesian modelling approach and increasing the number of simulated scenarios, for example. However, care must be taken as such transition may require an extensive computational time, careful consideration of model convergence criteria, and challenges in creating simple yet informative summary figures. We explicitly did not include the above in this study as our main objective was to create an illustrative, yet convincing example to convey the importance of combining a simulation–estimation exercise along with data fitting. The applicability of this approach is therefore not only limited to GSSM but to all other statistical models where simulation–estimation exercise can be performed.

**Take-home messages**. The main take-home of our work is two-fold. (i) We find that the estimated risk of large bias in population status could be substantially biased for data-limited species with high growth potential, high population variability, and weak density dependence, when using the GSSM to provide advice for management. (ii) More broadly, our findings demonstrate the importance of combining a simulation–estimation exercise along with data fitting to have a more accurate and robust view of the risks associated with management-relevant quantities and contribute towards a precautionary approach to management. Successful management of the living environment of our planet relies on models that tell the true story, hence we make a general call for improved risk assessment in conservation ecology.

## Methods

**Use of the GPDD**. GPDD is one of the largest collections of population time series available online and has been extensively used to study cross-taxa patterns in density dependence, extinction risks, population cycles, weather effect[18]. It contains more than 5000 time series of population abundance obtained from various forms of population surveys, and from many different taxa such as fish, insects, mammals, and birds. However, not all datasets are reliable. Thus, data were filtered out from the database using the same criteria as in ref. [18] i.e., harvest and non-index based data were removed, as well as data sampled at non-annual intervals and time series with less than 15 unique values (Supplementary Figure 6). The identities of the individual datasets analyzed are as follows: 1, 2, 3, 5, 6, 7, 8, 9, 10, 11, 12, 13, 14, 15, 16, 17, 18, 44, 45, 46, 47, 56, 57, 58, 59, 60, 61, 1090, 1093, 1097, 1101, 1102, 1104, 1106, 1109, 1111, 1112, 1115, 1116, 1159, 1160, 1163, 1166, 1170, 1174, 1181, 1185, 1188, 1189, 1207, 1235, 1239, 1314, 1315, 1316, 1317, 1318, 1319, 1320, 1321, 1322, 1323, 1324, 1325, 1327, 1328, 1329, 1331, 1333, 1334, 1336, 1337, 1339, 1340, 1341, 1342, 1343, 1347, 1348, 1349, 1350, 1351, 1352, 1353, 1354, 1355, 1356, 1357, 1358, 1359, 1360, 1361, 1362, 1363, 1364, 1365, 1366, 1367, 1368, 1369, 1377, 1402, 1403, 1405, 1505, 1507, 1508, 1516, 1517, 1522, 1523, 1524, 1526, 1536, 1602, 1612, 1613, 1618, 1626, 1628, 1633, 1660, 1664, 1667, 1669, 1670, 1671, 1674, 1783, 1792, 1826, 1828, 1829, 1830, 1831, 1857, 1858, 1860, 1865, 1866, 1868, 1869, 1870, 1875, 1876, 1877, 1878, 1879, 1880, 1881, 1882, 1883, 1884, 1885, 1886, 1887, 1888, 1889, 1893, 1894, 1927, 1941, 1949, 1964, 1965, 1966, 1968, 1970, 1971, 1973, 1974, 1976, 1978, 1981, 1982, 1983, 1984, 1986, 1987, 1988, 1990, 1991, 1992, 1993, 1994, 1998, 1999, 2003, 2004, 2005, 2006, 2007, 2008, 2009, 2011, 2012, 2013, 2015, 2016, 2017, 2018, 2019, 2020, 2024, 2025, 2026, 2027, 2028, 2031,

2032, 2033, 2034, 2066, 2096, 2097, 2098, 2721, 2722, 2726, 2732, 2733, 2736, 2737, 2757, 2758, 2759, 2771, 2774, 2775, 2778, 2780, 2781, 5019, 5020, 5024, 5026, 5029, 5032, 5033, 5034, 5035, 5036, 5037, 5038, 5039, 5040, 5044, 5045, 5047, 5049, 5054, 5055, 6057, 6059, 6061, 6144, 6522, 6527, 6528, 6529, 6530, 6531, 6532, 6533, 6534, 6535, 6536, 6537, 6538, 6541, 6542, 6547, 6549, 6550, 6552, 6553, 6554, 6555, 6558, 6561, 6564, 6565, 6566, 6567, 6568, 6570, 6571, 6582, 6588, 6589, 6590, 6633, 6634, 6657, 6673, 6674, 6676, 6677, 6678, 6688, 6713, 6714, 6715, 6764, 6765, 6770, 6863, 6864, 6865, 6866, 6867, 6868, 6869, 6870, 6871, 6872, 6873, 6874, 6875, 6876, 6877, 6878, 6879, 6880, 6881, 6882, 6883, 6884, 6885, 6886, 6887, 6888, 6889, 6890, 6891, 6892, 6893, 6894, 6895, 6896, 6897, 6898, 6899, 6900, 6901, 6902, 6903, 6904, 6905, 6906, 6907, 6908, 6909, 6910, 6911, 6912, 6913, 6914, 6915, 6916, 6917, 6918, 6919, 6920, 6921, 6922, 6923, 6924, 6925, 6926, 6927, 6928, 6929, 6930, 6931, 6932, 6933, 6934, 6935, 6936, 6937, 6938, 6939, 6940, 6941, 6942, 6943, 6944, 6945, 6946, 6947, 6948, 6949, 6950, 6951, 6953, 6955, 6956, 6957, 6958, 6959, 6960, 6961, 6962, 6963, 6964, 6965, 6966, 6967, 6968, 6970, 6971, 6972, 6973, 6974, 6975, 6976, 6977, 6978, 6982, 6983, 6985, 6986, 6987, 6989, 6990, 6991, 6992, 6993, 6994, 6995, 6996, 7022, 7025, 7041, 7042, 7048, 7051, 7059, 7060, 7061, 7067, 7088, 7089, 7091, 7092, 7093, 7094, 7096, 7115, 7119, 9185, 9186, 9187, 9188, 9191, 9192, 9200, 9207, 9211, 9232, 9245, 9247, 9270, 9271, 9272, 9273, 9276, 9277, 9279, 9280, 9294, 9308, 9330, 9331, 9339, 9344, 9347, 9355, 9357, 9370, 9371, 9372, 9373, 9374, 9375, 9376, 9377, 9378, 9379, 9381, 9391, 9393, 9395, 9400, 9436, 9437, 9438, 9439, 9440, 9441, 9442, 9443, 9444, 9445, 9446, 9447, 9460, 9463, 9506, 9507, 9513, 9515, 9519, 9641, 9685, 9686, 9688, 9689, 9690, 9691, 9692, 9693, 9694, 9695, 9696, 9698, 9699, 9700, 9701, 9702, 9703, 9704, 9705, 9706, 9707, 9708, 9709, 9710, 9711, 9712, 9713, 9714, 9715, 9717, 9718, 9719, 9720, 9721, 9722, 9723, 9724, 9725, 9726, 9727, 9728, 9729, 9730, 9731, 9732, 9733, 9734, 9736, 9737, 9738, 9739, 9740, 9741, 9793, 9794, 9795, 9796, 9797, 9832, 9835, 9836, 9837, 9855, 9856, 9857, 9858, 9859, 9860, 9861, 9862, 9863, 9864, 9865, 9866, 9867, 9868, 9869, 9870, 9871, 9872, 9873, 9874, 9875, 9876, 9877, 9878, 9879, 9880, 9881, 9882, 9883, 9884, 9885, 9886, 9887, 9888, 9893, 9894, 9896, 9897, 9898, 9899, 9901, 9902, 9903, 9904, 9907, 9947, 9948, 9949, 9950, 9951.

**The GSSM**. Our basic model expresses the change in log-population size, $\ln(N_t)$, over annual time step $t$, as a function of its growth potential i.e., the maximum rate that a population can increase from a time step to another, and a density-dependent effect on population increase, which accounts for processes such as competition, disease, and predation. These two processes are represented in the model by the parameter $\mu$ and $\rho$, respectively. On top of that, a stochastic term is added to the model to acknowledge our lack of understanding of the complex dynamics of natural systems. The latter is represented by the Gaussian distribution $\mathcal{N}$ with the variance term $\sigma^2_{proc}$.

$$\ln(N_t) \sim \mathcal{N}(\mu + \rho \ln(N_{t-1}), \sigma^2_{proc}) \qquad (1)$$

From the above population, sampling is performed to obtain an index of population abundance

$$\ln(O_t) \sim \mathcal{N}(\ln(N_t), \sigma^2_{obs}) \qquad (2)$$

The sampling of population abundance is often imperfect and comes with some level of uncertainty, which is again represented by a Gaussian distribution with variance term, $\sigma^2_{obs}$. We note that both $N_t$ and $O_t$ are log-normally distributed, thus are strictly positive.

**Simulation–estimation procedure**. To evaluate the performance of GSSM in estimating management-relevant parameters, we run a simulation–estimation procedure under a variety of scenarios representative of the population dynamics observed in the wildlife, across different taxa. The simulation–estimation experiment consisted of the following steps. (i) Create relevant simulation scenarios using GSSMs. A scenario was defined by a unique combination of model parameters (Supplementary Table 1). The scenarios were based on a literature review to determine the range of relevant parameters across taxa (Supplementary Table 1). This step generated a time series of population abundance and observations. (ii) We then fitted GSSMs to the generated data by using Template Model Builder (TMB)[23], a program that computes the marginal likelihood of the fixed effects and integrates over the random effects using Laplace approximation. The marginal likelihood was then maximized using the nonlinear maximization routine optim available in the R statistical environment[24] (version 3.5.2). Many other estimation approaches exist in the literature (both in frequentist and Bayesian frameworks) and have been used to fit GSSMs but studies showed that both Bayesian and frequentist approach produce biased parameter estimates and suffer from estimation problems[13]. (iii) We repeated this process many times until we got 50 results that converged (i.e., a successful convergence message code from optim and an invertible hessian matrix). Convergence failure rate during simulation–estimation varied between 0 and 0.7 (i.e., 70%) depending on the scenario (Supplementary Fig. 7). (iv) Then, we examined the bias in the estimated parameters as well as the estimated risks of biased population status (see section below). Bias was calculated as the absolute relative error rate: $\left|\frac{E(X)-X_{true}}{X_{true}}\right|$, with $X$

being the variable of interest. We run all possible combinations of parameters from Supplementary Table 1 to create a total of 135,000 scenarios.

**Fitting GSSMs to the GPDD**. In addition to the above simulation–estimation procedure, we also fitted GSSMs to the GPDD time series. Convergence was also examined based on a successful convergence message code from optim and an invertible hessian matrix. If the model did not converge, we re-iterated the model fitting process using a different parameter starting value (sampled randomly from an uniform distribution $\mu \sim U[0,\max(y)]$, $\rho \sim U[-1,1]$, $\ln(\sigma_{\text{obs}}) \sim U[-5,2]$, except $\ln(\sigma_{\text{proc}})$ that was fixed at 0 for all starting value combinations). If the model failed to converge after 1000 iterations, we flagged the time series as non-converged. Nine out of 627 GPDD time series (~1.4%) failed to converge thus were left out from the analysis. For these time series, no appropriate GSSM parameter values could be estimated with our methods thus, were not used together with the simulation–estimation results to back-calculate the "true" parameter range (see section below for definition and method detail). Additionally, we noted that for certain time series, GSSM estimated relatively large process error variance. To match the simulation–estimation exercise (which encompasses many realistic scenarios), we only plotted species for which the density-dependent parameter $\rho$ was estimated between $0 \le \rho \le 1$, and species with a total variability (coefficient of variation) between 0 and 2. CV was calculated in this study based on the total variance (sum of the observation and process error variance (in log scale)) and scaling it by the equilibrium population size (in log scale). See Eq. (1) and (2) for the meaning of the variables.

$$CV = \frac{\sqrt{\sigma_{\text{proc}}^2 + \sigma_{\text{obs}}^2}}{\mu/(1-\rho)} \quad (3)$$

**Risk evaluation**. Here, we focused on a risk measure with relevance for management i.e., the probability that the estimate of final-year population depletion is at least 50% biased i.e., an absolute relative error rate of 0.5. Population depletion estimate (i.e., the population size at the end of the time series compared to the population size at the start of the time series or another reference point) has often been used in fisheries to assess the status of fish populations and set catch limit (i.e., harvest quota) e.g., the so-called 40–10 rule set by the Pacific Fishery Management Council in the US West coast, or as a proposed management plan for European fisheries[25]. We referred to it as the risk of biased population status estimate. For each simulation scenario (i.e., specific life history), we calculated the probability (i.e., how often based on the extensive simulation–estimation runs that were conducted) that the final-year population estimate was at least 50% biased (i.e., the estimate is more than 50% off in either direction, compared to the truth used in the simulation). A species with high risk can have, for example, 80% chance of having at least a 50% bias in the final population status estimate. These results were subsequently used to create color-coded 2D maps, which summarize the risk of biased population status estimate with respect to important species life history characteristics (Figs 2, 3). Species falling in the orange-red zone have more than 60% chance of ≥50% bias, in the orange 30–60% change of ≥50% bias, and in bluish-green zone lower than 30% chance of ≥50% bias.

In addition to the above analysis of bias, we also used the results from extensive simulation–estimation (135,000 scenarios, 6.75 million datasets) to do a backward reasoning i.e., instead of looking at the range of parameter estimates (for each simulated dataset) that each scenario lead to, we looked at each simulated dataset (thus with a specific time series length) with its population parameter estimates and determined the combination of true population parameters (i.e., simulation scenario with the matching time series length), which could lead to such parameter estimates. In a sense, the approach is philosophically similar to the approximate Bayesian computation (ABC) but we do not make any direct use of the simulated data per se as opposed to the ABC, which require using a metric that compares the simulated data with the observed data. We will refer to it as "true" parameter back-calculation in this study. In order to do so, all parameters were binned into categories following the values chosen in the simulation analysis (Supplementary Table 1). As an example, a 25-year population time series ([15,30) years), with $\mu$, $\rho$, and CV estimated, respectively, between [0.3, 0.6], [0.55, 0.65], and [0.4, 0.6] might not only come from a population dynamic models with the same parameter combination but from a range of models with different parameter combinations: what we will refer to as back-calculated "true" parameter space. To help visualization, the back-calculated "true" parameter space was summarized in 2D using contour plots (the two-dimensional kernel density smoother function kde2d from the MASS package in R was used) over plots with the CV values on $x$-axis, $\rho$ values on the $y$-axis, and the growth category on each panel. This contour plots allow to visualize the global directionality of bias (i.e., the center of gravity of the back-calculated "true" parameter space) but also the inherent uncertainty associated with the study case (i.e., surface area of the back-calculated "true" parameter space).

We finally apply the approach to the GPDD data by linking the point estimates from the data and the time series length with the results from the simulation–estimation procedure to back-calculate the true parameter space.

**Reporting summary**. Further information on research design is available in the Nature Research Reporting Summary linked to this article.

## Data availability

The authors declare that all data supporting the findings of this study are available within the paper, the source data files, and from https://www.imperial.ac.uk/cpb/gpdd2/secure/login.aspx.

## Code availability

R code to reproduce the simulation–estimation as well as the data fitting to GPDD is included in the source data.

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

## Acknowledgements

We thank Nigel G. Yoccoz, Sean C. Anderson for comments on an earlier version of the manuscript. K.O. is funded by the Research Council of Norway through the Skagcore project (255675). Ø.L. is funded by the Research Council of Norway through the OIL-COM (255487) and FISHCOM (280467) projects.

## Author contributions

K.O. and Ø.L. designed the experiment, K.O. analyzed data, K.O., N.C.S., and Ø.L. were involved in results interpretation and drafting of the manuscript. All authors gave final approval for submission.

## Additional information

**Competing interests:** The authors declare no competing interests.

