## [Peer Review File · Nature Communications]

Reviewers' comments:

Reviewer #1 (Remarks to the Author):

Statistical models underpin many conservation efforts and management decisions. Uncertainties in management relevant parameters derived from these models can arise for a multitude of reasons including short time-series, high natural variability in population processes and measurement error. The authors illustrate this through fitting a simple population dynamics model to time series in the Global Population Dynamics database and an extensive simulation-estimation exercise. They find that estimates of depletion in abundance are often biased, and that simulation can be used to “back-calculate” a plausible range of true estimates of depletion. They also show these biases increase with the species' growth rate and population variability but decreases with the time series length and magnitude of density dependence.

I found the manuscript reasonably well written and the statistical and simulation modelling rigorous and novel, but I think the management relevance of the work is overstated and the discussion of risk superficial.

Throughout the manuscript the authors emphasize the management relevance of their work, but fail to mention that management decision are seldom based on the statistical examination of a time series of data alone, and so treating bias in estimates of depletion as a proxy for spurious management advice might be considered a bit of a stretch for some. For example, can the authors provide examples of where management decisions were actually based on estimates of depletion from the Gompertz model? I think that interpreting the interesting results of their work from the perspective of false positives and negatives in estimates of changes in abundance, as opposed to management advice, would be more appropriate in this case.

I found the consideration of risk, which the authors consider the absolute bias in estimates of depletion (change in abundance from beginning to end of the time series), to be superficial. For example, there was no discussion of types of risk or asymmetry in it. For example, overestimating depletion is less important than underestimating it from a biological perspective, but the opposite may be true in many cases from a socio-economic perspective. Figure 1 highlights that while estimates of depletion are highly uncertain (red) for some combinations of density dependence and process error, they are not biased in a particular direction (pie chart is equal parts light and dark grey), while for other parameter combinations (green, yellow areas) they are biased (overestimate depletion more often than underestimate it – at least according to L127, text beginning on L130 seems to contradict this) but in a biologically conservative direction. I would encourage the authors to consider the classification errors that arise in their analyses (i.e., false positives and false negatives) and what they might mean from both a conservation and socio economic risk perspective.

In a number of places there is a lack of clarity or detail which precludes a more complete understanding of the findings. For example, many figure captions (e.g., Fig 3) do not specify what magnitude of observation error is illustrated, or if the full range of simulated values considered is incorporated into the plot.

Lastly, I found it awkward that the methods were not written in the past tense.

Reviewer #2 (Remarks to the Author):

General comments

The paper risk assessment is novel, interesting and potentially important. However, as it currently stands there are critical areas where more information is needed to judge the paper's full capacity to contribute to the current literature (further information below).

The link to management scenarios as claimed (L49, L55, L66) is implicit in the model and therefore it may be overemphasised in the manuscript. The reader gets the impression that a list of management scenarios (L66) is created and applied to the modelling of the time series. However, it seems that this relates to varying population productivity, density dependence, depletion, time series length, population variability, and the ratio of observation to process error (Table S1 L433). Thus, the link to management may need to be toned down in the introduction and not mentioned in the results as it remains untested in the current modelling setup. The implications of uncertainty in population estimates (so far called management scenarios) for management can then be expanded upon in the discussion. If a more direct link with management is desired, then the different scenarios of population parameter combinations would need to be linked to broad management scenarios where for example certain management actions are connected to certain patterns in population dynamics (e.g. Di Fonzo et al. 2013, *Ecology and Evolution* 3, 2378).

A further general concern is the treatment of non-convergence and how this was treated. This needs clarification because the non-convergence could mean that there is substantial added uncertainty of choosing the right model, which is not discussed in the text but seems to be an important result. From this, it is then unclear whether the non-convergence cases were simply excluded when discussing the results of density dependence, bias of the population abundance and bias in growth estimates.

Specific comments

- L25 the link to biodiversity is a bit far fetched, the intro may better focus on wildlife species and population management and why that is important.
- L25 What does properly mean here? Do you mean sustainably? This is vague and needs clarification.
- L33-35 What is the link between uncertainty and risk? Please be more specific and explain what the conceptual thinking is behind this.
- L49 Please explain more clearly where and how the probability of final year depletion is used in management decisions. What is the link explicitly between the probability and a certain management decision, what kind of decisions follow and have there been any examples where this had detrimental outcomes for the population. It would be good to have more evidence that this is used, how it is used, and where it had serious implications for the status of populations.
- L87 Sampling of population abundance does not generally follow a Gaussian distribution because this would theoretically mean that counts could be negative. Also, most count data follow a Poisson distribution. Therefore, sampling may be done better with a Poisson distribution or a similar truncated distribution that does not produce negative numbers.
- L303 This might be more of an issue of explanation than modelling, but how is it possible that GSSM was first used to produce lots of time series of population abundance, and then GSSMs are fit to the generated data. Then L323, the GSSMs are fitted to the GDPP. Clarification is needed, especially because there is no space limitation in the supplementary material.
- L322 Were all possible combinations of parameters from table S1 run across all GDPP time series? L307/308 suggests that not all model parameters may be suitable for all taxa and species? Please clarify this contradiction.
- L329 Is it correct that 21 out of 627 GDPP failed to converge any of the 180,000 scenarios? It would be better to know how many of the scenarios failed. Also, how were the 21 time series treated and which ones were they? What was the reason that they didn't fit any of the 180,000 scenarios? It seems quite unlikely that not a single scenario fitted, given the wide range. Reasons for this should be understood. Reading further on, the failure rate, given in Fig S2, is substantial, over 60% in many

cases, would this not indicate that the GSSMS model with density dependence (especially at lower rates) is not a suitable model? Knappe and Valpine (2012) argue that density dependence is hard to detect and measure in the GPDD data, and that may limit the current study.

- L331 How was the non-convergence treated when evaluating risk?

Reviewer #3 (Remarks to the Author):

I thank authors for their effort in devising this manuscript. The premise of this paper is highly relevant to the current state and future direction of ecological sciences. Simulations as a supporting mechanism for real data analysis and incorporating uncertainty in the presentation of results are topics of high importance and this manuscript makes significant contributions to these methodological advances. Patterns found with density dependence, growth rate, and process variance, and their relation to decision making in conservation biology are also good additions to the literature. I have, however, two major criticisms about the way results are presented. I do not suggest a re-analysis except in one case (and even so some additional arguments might suffice), but re-writing parts of the background and results sections is necessary.

The first criticism is about the back calculation of true parameter range, as explained in Fig 1. This is an interesting concept, and to my knowledge is a novel idea. The authors, however, can make some improvements in its presentation. At first glance, "true parameter range" does seem like a regular confidence interval. The difference is that it is actually a distribution of the parameter itself. Frequentist methods assume true parameters are fixed, they can't have distributions, so in that case it can be considered to be more similar to a Bayesian credible interval or posterior distribution. It is not, of course, obtained in a Bayesian framework with a proper MCMC algorithm, so it is also not a posterior distribution. The distribution is obtained via extensive population simulations, and it corresponds to a true parameter distribution conditional on an estimate for that parameter. The estimation process is frequentist (maximum likelihood), but the distribution in question is some distribution of the true parameter obtained with simulations. It has close ties to both confidence intervals and credible intervals but it is neither. The method itself can even be considered similar to Approximate Bayesian Computation (APC), at least on philosophical grounds. I think the authors should discuss this nuance in the manuscript in more detail. It might be better to explicitly name the method rather than just saying "back-calculation".

The major issue is that the true parameter range depends solely on the expected value of the parameter in question. While it is useful to calculate bias with a point estimate, the motivation behind the true parameter range is the incorporation of uncertainty. Here the uncertainty in parameter estimation is completely ignored. So two different models with the same point estimates but with different confidence intervals would still lead to the same true parameter range. While authors acknowledge incorporating parameter uncertainty as a future goal, I don't see why it was not included in this study in the first place. It has the potential to substantially change the results (main take home message can actually change), and if authors insist on its exclusion from this manuscript they should provide a better argument for their decision.

The second criticism is about the decision to restrict the simulations to a single model type, which is Gompertz State Space Models (GSSM). This wouldn't have been an issue if the goal was to only explore the inherent biases with GSSM. However, the authors go on to provide a "most plausible parameter value" depending on these simulations, and also generalize their results for conservation management by calculating an index, again using these simulations. The authors present no evidence that 627 population time series data is the result of a process similar to GSSM. Concurrently, the bias

structure might change if data simulations were done with a different model than GSSM but it's analysis was still done with GSSM. The results reported, then, is conditional on the fact that GSSM is a good approximation of the natural processes that lead to all of the 627 population time series. This is a big assumption and is not mentioned in the manuscript. Authors reference GSSM to be widely used in the ecological literature but none of the cited papers actually compare GSSM to other model types that may include explicit parameters for inter-specific competition, predation, or lag effects in density dependence, or that may simply have a different structure (e.g. theta-logistic).

I'm not suggesting to add other model types to the current manuscript, although I think this a good area to explore and can be mentioned as a future goal. The presentation of the results, however, should change and it should be made abundantly clear that every result found with the 627 time series data comes with the assumption that GSSM is a good approximation of the natural processes that led to these time series. There could be many (or maybe only some) species among the 627 that GSSM is not a good fit. So patterns found with growth rate and density dependence strength (Figures 2 and 3) present only a piece of the evidence with a significant assumption. The more interesting and novel point is the applicability and generality of the presented framework and not GSSM and its accompanying patterns. The clear message of the manuscript for me is: If we have a species with a time series data, and if we have a simple model that we can show is a good candidate for natural processes that generates this time series data, then we can use simulations to incorporate uncertainty and to correct inherent model biases in decision making for species conservation. The assumptions of this framework and its applicability to other statistical models should be more clearly stated.

I also find Figures 2 and 3 hard to understand. The definition of risk zones and point estimates are clear. The arrows, however, are a bit confusing. The arrows point the most plausible true parameter estimate. Because this parameter estimate is now "true", it should not point to another risk zone. If it is true, there is no risk of bias. It seems as if though, for some species bias correction can make things worse by moving the estimate to a higher risk zone. The risk zone of a point estimate and it's bias correction with simulations are two different processes and belong to different figures.

Some additional suggestions:

- It might be a good idea to break the sentence starting at line 49 to multiple sentences. I read it several times to understand the definition of the index.
- There should be iii) at line 316, not ii).
- The rows in Figure S5 are same with the previous graphs . Why are they named differently?

Reviewer #1 (Remarks to the Author):

Statistical models underpin many conservation efforts and management decisions. Uncertainties
in management relevant parameters derived from these models can arise for a multitude of
reasons including short time-series, high natural variability in population processes and
measurement error. The authors illustrate this through fitting a simple population dynamics
model to time series in the Global Population Dynamics database and an extensive simulation-
estimation exercise. They find that estimates of depletion in abundance are often biased, and that
simulation can be used to “back-calculate” a plausible range of true estimates of depletion. They
also show these biases increase with the species’ growth rate and population variability but
decreases with the time series length and magnitude of density dependence.

I found the manuscript reasonably well written and the statistical and simulation modelling
rigorous and novel, but I think the management relevance of the work is overstated and the
discussion of risk superficial.

→ We very much appreciate your comments. We have now downscaled the management
relevance of our work (both in the introduction and method) and mostly talk about it in the
discussion. In practice, the wording “risk of spurious management advice” was replaced
throughout the manuscript by “risk of biased population status estimate” and many other re-
wording was put into place. See below for further details.

Throughout the manuscript the authors emphasize the management relevance of their work, but
fail to mention that management decision are seldom based on the statistical examination of a
time series of data alone, and so treating bias in estimates of depletion as a proxy for spurious
management advice might be considered a bit of a stretch for some. For example, can the authors
provide examples of where management decisions were actually based on estimates of depletion
from the Gompertz model?

→ Thank you for this very important comment of yours: We did not intend to overstate the
management relevance of our work. To alleviate this impression, we have downscaled the focus
on the management relevance, but we also specifically mention one example of a management
body that use these depletion estimates directly to provide management advice.

The text now reads: “Here we focus on a risk measure with management relevance: the
probability that the estimate of final year population depletion is at least 50% biased i.e. an
absolute relative error rate of 0.5. Population depletion estimate – estimate of population size at
the end of the time series compared to the population size at the start of the time series – is often
used to assess population status in fisheries and set catch limit accordingly i.e. if this ratio is
below a given limit, no harvest is allowed, but when this ratio is high, a higher catch is allowed.
The so-called “40-10 rule” used in the US West coast by the Pacific Fishery Management
Council is an example. We refer to this risk measure as “risk of biased population status
estimate”. Through an extensive simulation-estimation procedure (> 5 million datasets were
generated. Fig. 1 and Supplementary Material), we quantify the risk of biased population status
estimate for 627 time series of population abundance from the Global Population Dynamics
Database, covering four taxonomic groups¹⁶.”

[Redacted]

Figure 1. The 40-10 control rule used by the Pacific Fishery Management Council (PFMC).

Figure taken from: Punt and Ralston 2007. A management strategy evaluation of rebuilding revision rules for overfished rockfish stocks

The figure above illustrates the 40-10 rule i.e. when stock size falls below 10% of B_0 (i.e. the virgin population biomass), then no harvest is allowed. When the stock size is between 10 and 40% of B_0 , then catch is linearly reduced from the that when population is at 40% of B_0 (i.e biomass x F_{msy}). Above 40% of B_0 , catch is set equal to biomass x F_{msy} .

I think that interpreting the interesting results of their work from the perspective of false positives and negatives in estimates of changes in abundance, as opposed to management advice, would be more appropriate in this case.

I found the consideration of risk, which the authors consider the absolute bias in estimates of depletion (change in abundance from beginning to end of the time series), to be superficial. For example, there was no discussion of types of risk or asymmetry in it. For example, overestimating depletion is less important than underestimating it from a biological perspective, but the opposite may be true in many cases from a socio-economic perspective. Figure 1 highlights that while estimates of depletion are highly uncertain (red) for some combinations of density dependence and process error, they are not biased in a particular direction (pie chart is equal parts light and dark grey), while for other parameter combinations (green, yellow areas) they are biased (overestimate depletion more often than underestimate it – at least according to L127, text beginning on L130 seems to contradict this) but in a biologically conservative direction. I would encourage the authors to consider

the classification errors that arise in their analyses (i.e., false positives and false negatives) and what they might mean from both a conservation and socio economic risk perspective.

→ Thank you for your comments and suggestions: We have now downscaled the management relevance of our work and also talk about false-positives and negatives when examining the risk, in addition to their possible implications.

The text now reads: “Furthermore, there were more false-positives in risk estimates (i.e. thinking that population is healthier than it actually is by at least 50%) for species in the lower risk

category (i.e. species with lower growth rate, lower population variability, and stronger density
 dependence) but the proportion of false-positives decreases with the species risk category (Fig.
 2A-E, S6-7). This suggests that estimates of population depletion for species categorized in the
 lower risk zone have a low chance (<30%) of being very biased (>50% bias) but when they are,
 they have close to 75% chance of overestimating population status. An overestimation of the
 population status may lead to an elevated risk of population reduction (if population depletion
 estimate is used for setting species harvest limit e.g. the so-called “40-10 rule” in fisheries) with
 negative consequences for conservation. Conversely, an underestimation of population status
 may lead to underestimation of the harvest potential of the population with negative socio-
 economic consequences

In a number of places there is a lack of clarity or detail which precludes a more complete
 understanding of the findings. For example, many figure captions (e.g., Fig 3) do not specify
 what magnitude of observation error is illustrated, or if the full range of simulated values
 considered is incorporated into the plot.

→ Sorry for the unclarity: We have now extensively expanded the supplementary material to
 include detailed information on the method. Moreover, we have extended the figure captions to
 clarify our approach.

Lastly, I found it awkward that the methods were not written in the past tense.

→ We agree that it is better to write the methods in the past tense, hence we have now changed
 the tense in the materials and methods (where applicable).

**Reviewer #2 (Remarks to the Author):**

General comments

The paper risk assessment is novel, interesting and potentially important. However, as it
 currently stands there are critical areas where more information is needed to judge the paper’s
 full capacity to contribute to the current literature (further information below).

The link to management scenarios as claimed (L49, L55, L66) is implicit in the model and
 therefore it may be overemphasised in the manuscript. The reader gets the impression that a list
 of management scenarios (L66) is created and applied to the modelling of the time series.
 However, it seems that this relates to varying population productivity, density dependence,
 depletion, time series length, population variability, and the ratio of observation to process error
 (Table S1 L433). Thus, the link to management may need to be toned down in the introduction and
 not mentioned in the results as it remains untested in the current modelling setup.

The implications of uncertainty in population estimates (so far called management scenarios) for
 management can then be expanded upon in the discussion. If a more direct link with
 management is desired, then the different scenarios of population parameter combinations would
 need to be linked to broad management scenarios where for example certain management actions

are connected to certain patterns in population dynamics (e.g. Di Fonzo et al. 2013, Ecology and
Evolution 3, 2378).

→ Thank you for your advice: We also received a similar comment from reviewer 1 (see above).
We have now toned down the link to management throughout the text including the introduction,
methods, and results. In practice, the wording “risk of spurious management advice” was
replaced throughout by “risk of biased population status estimate” and we rather talk about
“management relevant parameter” instead of “management scenario”. In addition to the above,
we have also performed many other changes throughout the manuscript, to tone down the focus
on the management scenarios.

As an example, we have reworded “management scenarios” (in the SI) to “variety of scenarios
representative of the population dynamics observed in the wildlife, across different taxa”.

Additionally, in the methods we state: “To evaluate the performance of GSSM in estimating
management relevant parameters, we run a simulation-estimation procedure under a variety of
scenarios representative of the population dynamics observed in the wildlife, across different
taxa.”

Moreover, in the introduction, we have changed the text to: “Here we focus on a risk measure
with management relevance: the probability that the estimate of final year population depletion is
at least 50% biased i.e. an absolute relative error rate of 0.5. Population depletion estimate –
estimate of population size at the end of the time series compared to the population size at the
start of the time series – is often used to assess population status in fisheries and set catch limit
accordingly i.e. if this ratio is too low, no harvest is allowed, but when this ratio is high, a higher
harvest rate is allowed. The so-called “40-10 rule” used in the US West coast by the Pacific
Fishery Management Council is an example. We refer to this risk measure as “risk of biased
population status estimate”. Through an extensive simulation-estimation procedure (> 5 million
datasets were generated. Fig. 1 and Supplementary Material), we quantify the risk of biased
population status estimate for 627 time series of population abundance from the Global
Population Dynamics Database, covering four taxonomic groups¹⁶.”

A further general concern is the treatment of non-convergence and how this was treated. This
needs clarification because the non-convergence could mean that there is substantial added
uncertainty of choosing the right model, which is not discussed in the text but seems to be an
important result. From this, it is then unclear whether the non-convergence cases were simply
excluded when discussing the results of density dependence, bias of the population abundance
and bias in growth estimates.

→ We are sorry for the confusion we created and the lack of clarity in our text: We have now
expanded the SI to be more explicit about how and why we treated non-convergence in this
analysis.

When we describe the fitting of Gompertz state-space models (GSSMs) to abundance timeseries
from the Global Population Dynamics Database (GPDD) in the SI, we have now written: “In
addition to the above simulation-estimation procedure, we also fitted GSSMs to the GPDD time
series. Convergence was also examined based on a successful convergence message code from
*optim* and an invertible hessian matrix. If the model did not converge, we re-iterated the model
fitting process using a different parameter starting value (sampled randomly from an uniform

distribution $\mu \sim U[0, \max(y)]$, $\rho \sim U[-1, 1]$, $\ln(\sigma_{obs}) \sim U[-5, 2]$ except $\ln(\sigma_{proc})$ that was fixed at 0
 for all starting value combinations). If the model failed to converge after 1000 iterations, we
 flagged the time series as non-converged. 9 out of 627 GPDD time series (~1.4%) failed to
 converge thus were left out from the analysis. For these time series, no appropriate GSSM
 parameter values could be estimated with our method thus, were not used together with the
 simulation-estimation results to back-calculate the true parameter range. Additionally, we noted
 that for certain time series, GSSM estimated relatively large process error variance. To match the
 simulation-estimation exercise (which encompasses many realistic scenarios), we only plotted
 species for which the density dependent parameter ρ was estimated between $0 \leq \rho \leq 1$, and species
 with a total variability (coefficient of variation) between 0 and 2. CV was calculated in this study
 based on the total variance (sum of the observation and process error variance (in log scale)) and
 scaling it by the equilibrium population size (in log scale). See equations 1 and 2 for the meaning
 of the variables.

$$CV = \frac{\sqrt{\sigma_{proc}^2 + \sigma_{obs}^2}}{\mu / (1 - \rho)}$$

“

Specific comments

- L25 the link to biodiversity is a bit far fetched, the intro may better focus on wildlife species
 and population management and why that is important.

→ We have re-written the introduction to focus more on wildlife conservation and population
 management and we no longer discuss biodiversity.

The text now reads: “One of the main challenges and goals in conservation ecology is to
 sustainably manage the integrity of the ecosystem. A healthy ecosystem provides numerous
 services to the human population including food, climate regulation, disease regulation, nutrient
 cycling, and cultural experience¹. In order to manage an ecosystem sustainably with its wildlife
 and surrounding environment, sound management decisions based on the knowledge of the
 structure and dynamics of the ecosystem are required²”

- L25 What does properly mean here? Do you mean sustainably? This is vague and needs
 clarification.

→ We meant “sustainably” and have changed the text accordingly.

- L33-35 What is the link between uncertainty and risk? Please be more specific and explain
 what the conceptual thinking is behind this.

→ We have now reworded the sentence to be more explicit. We have now written:

“Statistical models have often served as a workhorse behind many conservation or management
 decisions by providing important estimates of population status. These include examples of
 IUCN red list definition³, endangered species listing in the US or Canada⁴, fisheries catch limits
 estimation⁵, and more. However, models are mere simplifications of the complex dynamics of
 the natural system⁶. Therefore, appropriate consideration of uncertainty is crucial to avoid
 fallacious interpretation of ecosystem functioning^{7,8}. Sources of uncertainty represent, for

example, our lack of knowledge on the mechanisms that govern the population dynamics or
 errors associated with measurements of the population. In this regard, it has become increasing
 popular and important to quantify risks (used in this study as the probability that something
 harmful to the society might happen e.g. risk of overexploitation, risk of population extinction, or
 risk of biased population status estimate) in population and conservation ecology^{9,10} and many
 institutions as well as regulations now require a precautionary approach to conservation and
 management¹¹. “

- L49 Please explain more clearly where and how the probability of final year depletion is used
 in management decisions. What is the link explicitly between the probability and a certain
 management decision, what kind of decisions follow and have there been any examples where
 this had detrimental outcomes for the population. It would be good to have more evidence that
 this used, how it is used, and where it had serious implications for the status of populations.

→ Similar issue was raised by the other reviewer. Therefore, we have now toned down the
 management relevance of our work. Coming back to the topic of Gompertz model and use of
 depletion estimate for management decision: in fisheries, a family of models known as surplus
 production models or biomass dynamics models (e.g. Schaefer, Fox, or Pella-Tomlinson models)
 - which are in the same family as the Gompertz model - have all been used from many decades
 (see for example Hilborn and Walters 1992 “Quantitative Fisheries Stock Assessment Choice,
 Dynamics and Uncertainty” for reference herein) and are kept being used in many jurisdictions
 (where fishery data are scarce) to assess the status of the fish stock. On a side note, the Fox
 biomass dynamics model (Fox 1970. An exponential surplus-yield model for optimizing
 exploited fish populations) is a Taylor series approximation of the Gompertz model. Additionally,
 management advice in fisheries are based on some harvest control rule (HCR, i.e. rules that
 dictate when and how to set fisheries catch quota) and the HCR is often based on some
 measurement of the current level of biomass compared to a reference level e.g. the so-called “40-
 10 rule” applied by the Pacific Fishery Management Council (who is the management body of
 the US West coast fishery i.e. the states of Washington, Oregon, and California) determines the
 harvest rate (thus the quota) that is allowed to catch for each species by assessing the final
 population depletion value. If the population is above 40% of B_0 , then a harvest rate equivalent
 to FMSY (harvest that leads to the maximum sustainable yield in the equilibrium condition) is
 applied. If the population falls between 10% and 40% of B_0 , then catch is linearly decreased
 from the level that corresponds to Fmsy until no fishing is allowed when the population reaches
 below 10% of B_0 . Additional precautionary approach is also added on top of this to account for
 scientific uncertainty (See figure in the response to reviewer 1).

We have now written the following in the text:

“Here we focus on a risk measure with management relevance: the probability that the estimate
 of final year population depletion is at least 50% biased i.e. an absolute relative error rate of 0.5.
 Population depletion estimate – estimate of population size at the end of the time series
 compared to the population size at the start of the time series – is often used to assess population
 status in fisheries and set catch limit accordingly i.e. if this ratio is below a given limit, no
 harvest is allowed, but when this ratio is high, a higher catch is allowed. The so-called “40-10

rule” used in the US West coast by the Pacific Fishery Management Council is an example. We
 refer to this risk measure as “risk of biased population status estimate”. Through an extensive
 simulation-estimation procedure (> 5 million datasets were generated. Fig. 1 and Supplementary
 Material), we quantify the risk of biased population status estimate for 627 time series of
 population abundance from the Global Population Dynamics Database, covering four taxonomic
 groups¹⁶.”

- L87 Sampling of population abundance does not generally follow a Gaussian distribution
 because this would theoretically mean that counts could be negative. Also, most count data
 follow a Poisson distribution. Therefore, sampling may be done better with a Poisson distribution
 or a similar truncated distribution that does not produce negative numbers.

→ Sorry for the lack of clarity: We applied a gaussian distribution to the log(abundance).

Therefore, the abundance is positively distributed (thus in real scale, abundance is log-normally
 distributed), and there is no negative numbers. Moreover, such modeling framework and
 distribution assumptions for GSSM has a long-standing in ecological literature.

We have now clarified this in the text: “We note that both N_t and O_t are log-normally distributed,
 thus are strictly positive”

- L303 This might be more of an issue of explanation than modelling, but how is it possible that
 GSSM was first used to produce lots of time series of population abundance, and then GSSMs
 are fit to the generated data. Then L323, the GSSMs are fitted to the GDPP. Clarification is
 needed, especially because there is no space limitation in the supplementary material.

→ Sorry for the lack of clarity: We have now rewritten parts of the method section to better
 explain what we did (as suggested, there is no space limitation in the supplementary material,
 hence we can expand to provide a more detailed but clear explanation).

We suggest a two-step approach:

**Step 1** was to conduct a simulation-estimation experiment. In this step, we used GSSM with a
 certain parameter combination (what we called scenario in the text) to generate time series of
 population abundance. This is the step we refer to as “simulation”. In practice, it means
 generating population abundance trajectories by sampling process and observation noise and
 combining it with the GSSM parameters. Next, we used this generated index of abundance as
 data to fit the GSSM and obtain some parameter estimates. This is the “estimation” phase. We
 repeat this process hundreds of time for each scenario and this whole process is called
 “simulation-estimation”.

**Step 2** was to fit GSSM to the GPDD data to obtain GSSM parameter estimates. However,
 thanks to the “simulation-estimation” in step 1, we now know that these parameter estimates
 could have been obtained by various parameters combinations. Hence, we tried to identify the
 region of “true” parameter space from which these estimates could have been derived from
 (therefore the requirement of extensive simulation-estimation).

The text now reads: “To evaluate the performance of GSSM in estimating management relevant
 parameters, we run a simulation-estimation procedure under a variety of scenarios representative
 of the population dynamics observed in the wildlife, across different taxa. The simulation-

estimation experiment consisted of the following steps. i) Create relevant simulation scenarios
 using GSSMs. A scenario was defined by a unique combination of model parameters (Table S1).
 The scenarios were based on a literature review to determine the range of relevant parameters
 across taxa (Table S1). This step generated a time series of population abundance and
 observations. ii) We then fitted GSSMs to the generated data by using Template Model Builder
 (TMB)²³, a program that computes the marginal likelihood of the fixed effects and integrates
 over the random effects using Laplace approximation. The marginal likelihood was then
 maximized using the nonlinear maximization routine *optim* available in the R statistical
 environment²⁴. Many other estimation approaches exist in the literature (both in frequentist and
 Bayesian frameworks) and have been used to fit GSSMs¹³ but showed that both Bayesian and
 frequentist approach produce biased parameter estimates and suffer from estimation problems.
 iii) We repeated this process many times until we got 50 results that converged (i.e. a successful
 convergence message code from *optim* and an invertible hessian matrix). Convergence failure
 rate during simulation-estimation varied between 0 and 0.7 (i.e. 70%) depending on the scenario
 (Fig. S2). iv) Then, we examined the bias in the estimated parameters as well as the estimated
 risks of biased population status (see section below). Bias was calculated as the absolute relative
 error rate: $\left| \frac{E(X) - X_{true}}{X_{true}} \right|$, with X being the variable of interest. We run all possible combinations of
 parameters from Table S1 to create a total of 135.000 scenarios.”

See also answer to the comment below.

- L322 Were all possible combinations of parameters from table S1 run across all GDPP time
 series? L307/308 suggests that not all model parameters may be suitable for all taxa and species?
 Please clarify this contradiction.

→ Thank you for noticing the lack of description in the text: We have now clarified this in the
 method section. As you suspected, not all parameter combinations were used to derive the “true
 parameter space” for each estimate from the GPDD time series. The main separation (as you can
 see in the figure 2, S6-7) happened with the time series length (categorized by short: 11-20 years
 (using the results from 15-years scenario), medium: 21-30 (using results from 25 years scenario),
 and large: 35-65 (using results from 50 years scenario)), the estimated species growth rate
 (categorized into very slow-growing (growth parameter estimated between 0-0.3), slow-growing
 (growth parameter estimated between 0.3-0.7), medium-growing (growth parameter estimated
 between 0.7-1.1), fast-growing (growth parameter estimated between 1.1-1.5), and very fast-
 growing (growth parameter estimated above 1.5), the estimated level of density dependence ρ ,
 and the estimated total variability in the study system CV.

“The text now reads: “In addition to the above analysis of bias, we also used the results from
 extensive simulation-estimation (135.000 scenarios) to do a backward reasoning i.e. instead of
 looking at the range of parameter estimates (for each simulated dataset) that each scenario lead to,
 we looked at each simulated dataset (thus with a specific time series length) with its population
 parameter estimates and determined the combination of true population parameters (i.e.
 simulation scenario with the matching time series length) which could lead to such parameter
 estimates. We will refer to it as “true parameter back-calculation” in this study. In order to do so,

all parameters were binned into categories following the values chosen in the simulation analysis
 (Table S1)). As an example, a 25-year population time series (for time series length of 21 to 30
 331 years), with μ , ρ , and CV estimated respectively between [0.3, 0.6], [0.55, 0.65], and [0.4, 0.6]
 might not only come from a population dynamic models with the same parameter combination
 but from a range of models with different parameter combinations: what we will refer to as
 “back-calculated true parameter space”. To help visualization, the back-calculated true parameter
 space was summarized in 2D using contour plots (the two-dimensional kernel density smoother
 function *kde2d* from the *MASS* package in R was used) over plots with the CV values on x-axis,
 ρ values on the y-axis, and the growth category on each panel. This contour plots allow to
 visualize the global directionality of bias (i.e. the center of gravity of the back-calculated true
 parameter space) but also the inherent uncertainty associated with the study case (i.e. surface
 area of the back-calculated parameter space).

We finally apply the approach to the GPDD data by linking the point estimates from the data and
 the time series length with the results from the simulation-estimation procedure to back-calculate
 the true parameter space.”

- L329 Is it correct that 21 out of 627 GDPP failed to converge any of the 180.000 scenarios? It
 would be better to know how many of the scenarios failed. Also, how were the 21 time series
 treated and which ones were they? What was the reason that they didn’t fit any of the 180.000
 scenarios? It is seems quite unlikely that not a single scenario fitted, given the wide range.
 Reasons for this should be understood. Reading further on, the failure rate, given in Fig S2, is
 substantial, over 60% in many cases, would this not indicate that the GSSMS model with density
 dependence (especially at lower rates) is not a suitable model? Knape and Valpine (2012) argue
 that density dependence is hard to detect and measure in the GPDD data, and that may limit the
 current study.

→ We acknowledge that we were unclear about this in the previous version of the manuscript.
 When we say that 21 out of 627 GPDD failed to converge, it means that we could not run GSSM
 on 21 species in GPDD: the model did not convergence for these datasets, hence we could not
 obtain any model estimates. Moreover, after increasing the number of iterations with different
 starting parameter values to 1000 (as opposed to 50 initially), the number of convergence failure
 decreased from 21 to 9. Because these species do not have any point estimates (point estimates
 are NAs), they are not plotted in the figures, thus, we cannot back-calculate the true parameter
 range based on the simulation-estimation results. We have now clarified this in the text (please
 see our response in L151-174 in this document). Moreover, we have further clarified the notion
 of convergence in the text as it appears in two occasions. One for the simulation-estimation step
 (and this one means how hard it is to fit a GSSM to certain scenario), and one for fitting GSSM
 to the real data (please see our response in L267-306 in this document).

- L331 How was the non-convergence treated when evaluating risk?

→ Sorry for our lack of clarity: As mentioned in our reply above, we have now clarified this
 notion of “non-convergence” in the text. This notion appears in two occasions in our study: one
 for the simulation-estimation step, and one for fitting GSSM to the real data.
 In the simulation-estimation, the amount of non-convergence indicates the difficulty one has to
 fit a GSSM to a certain type of scenario (e.g. with high observation to process error). Hence, we
 just eliminated the results from non-converged cases, and kept running new iterations until we
 reach a total of 50 iterations (we increased it from 20 in the previous version of the manuscript)
 that converged. The main objective of the simulation-estimation is to reproduce enough
 iterations (here 50) to cover the range of estimated values one can encounter which form the
 basis of risk evaluation. Additionally, we tested running 100 iterations for one scenario but it did
 not qualitatively change the result.
 In the case of real data fitting, this simply means that no GSSM parameter estimates could be
 obtained, thus we simply excluded from the analysis as these could not be plotted in the figures.

 In response to this comment (and similar comments above), we have changed the text to:
 “We repeated this process many times until we got 50 results that converged (i.e. a successful
 convergence message code from *optim* and an invertible hessian matrix).”
 And for real data fitting:
 “If the model failed to converge after 1000 iterations, we flagged the time series as non-
 converged. 9 out of 627 GPDD time series (~1.4%) failed to converge thus were left out from the
 analysis. For these time series, no appropriate GSSM parameter values could be estimated thus
 could not be used together with the simulation-estimation results to back-calculate the true
 parameter range.”

 **Reviewer #3 (Remarks to the Author):**

 I thank authors for their effort in devising this manuscript. The premise of this paper is highly
 relevant to the current state and future direction of ecological sciences. Simulations as a
 supporting mechanism for real data analysis and incorporating uncertainty in the presentation of
 results are topics of high importance and this manuscript makes significant contributions to these
 methodological advances. Patterns found with density dependence, growth rate, and process
 variance, and their relation to decision making in conservation biology are also good additions to
 the literature. I have, however, two major criticisms about the way results are presented. I do not
 suggest a re-analysis except in one case (and even so some additional arguments might suffice),
 but re-writing parts of the background and results sections is necessary.

 The first criticism is about the back calculation of true parameter range, as explained in Fig 1.
 This is an interesting concept, and to my knowledge is a novel idea. The authors, however, can
 make some improvements in its presentation. At first glance, “true parameter range” does seem
 like a regular confidence interval. The difference is that it is actually a distribution of the
 parameter itself. Frequentist methods assume true parameters are fixed, they can’t have
 distributions, so in that case it can be considered to be more similar to a Bayesian credible
 interval or posterior distribution. It is not, of course, obtained in a Bayesian framework with a
 proper MCMC algorithm, so it is also not a posterior distribution. The distribution is obtained via

extensive population simulations, and it corresponds to a true parameter distribution conditional
 on an estimate for that parameter. The estimation process is frequentist (maximum likelihood),
 but the distribution in question is some distribution of the true parameter obtained with
 simulations. It has close ties to both confidence intervals and credible intervals but it is neither.
 The method itself can even be considered similar to Approximate Bayesian Computation (APC),
 at least on philosophical grounds. I think the authors should discuss this nuances in the
 manuscript in more detail. It might be better to explicitly name the method rather than just saying
 “back-calculation”.

→ Thank you very much for the insightful comment: We have expanded the description of the
 “true parameter back-calculation” to be more explicit about this idea and have also made few
 modifications to Figure 1 and its caption (to be more explicit). We also added a footnote on how
 our approach is philosophically similar to ABC (albeit different).

 The text now reads: “In addition to the above analysis of bias, we also used the results from
 extensive simulation-estimation (135.000 scenarios) to do a backward reasoning i.e. instead of
 looking at the range of parameter estimates (for each simulated dataset) that each scenario lead to,
 we looked at each simulated dataset (thus with a specific time series length) with its population
 parameter estimates and determined the combination of true population parameters (i.e.
 simulation scenario with the matching time series length) which could lead to such parameter
 estimates¹. We will refer to it as “true parameter back-calculation” in this study.”

 As a footnote, we included:

“¹In a sense, the approach is philosophically similar to the Approximate Bayesian computation
 (ABC) but we do not make any direct use of the simulated data per se as opposed to the ABC
 (which require using a metric that compares the simulated data with the observed data) nor are
 we working in Bayesian framework. Hence this approach is not an ABC.”

 The major issue is that the true parameter range depends solely on the expected value of the
 parameter in question. While it is useful to calculate bias with a point estimate, the motivation
 behind the true parameter range is the incorporation of uncertainty. Here the uncertainty in
 parameter estimation is completely ignored. So two different models with the same point
 estimates but with different confidence intervals would still lead to the same true parameter
 range. While authors acknowledge incorporating parameter uncertainty as a future goal, I don’t
 see why it was not included in this study in the first place. It has the potential to substantially
 change the results (main take home message can actually change), and if authors insist on its
 exclusion from this manuscript they should provide a better argument for their decision.

→ Thank you for the insightful comment: We agree with the reviewer that incorporating both
 parameter estimation error in addition to this “true parameter range” uncertainty would be good.
 Indeed, we actually tried to run the models in a Bayesian framework at an early stage of this
 work, but we quickly realized the limitations associated with such an approach. Bayesian model
 output is only valid if the model has converged. But in order to perform a proper convergence
 test, one cannot simply look at the R-hat value which is the simplest metric one can extract from
 a Bayesian model run (although there exist a few papers in the literature which solely relied on
 this metric to evaluate model convergence) but is not a sufficient condition for model

convergence. At least a few different convergence tests need to be performed in addition to
getting the R-hat value to perform analysis e.g. visual exploration of the trace plot to verify the
chain mixing, making sure that the burn-in period was removed. In addition to this technical
challenge, one will have the challenge to creating simple yet informative figures. Finally, and
most importantly, the main purpose of this study was more to illustrate the concept (while future
improvement on the approach is possible), and convince the scientific community about the
importance of combining simulation-estimation with data fitting.

However, we tried including parameter estimation uncertainty for one specific case as an
illustrative example (Fig. S8). Doing so only enlarged parameter uncertainty (the surface area
covered by the contour plot) for this specific case.

Importantly, we also noticed that the text in the previous version was misleading. When we
wrote “Although we did not include such uncertainty in this study, the main take-home messages
remain unchanged”, we were ONLY referring to the importance of combining simulation-
estimation along with data fitting, not on the results of the GSSM and GPDD per se.

To clarify these points, we have rewritten the text to: “Concomitantly, future studies should also
consider other sources of uncertainty such as parameter estimation uncertainty (see Fig. S8 for
illustrative example) or model types uncertainty (e.g. using a theta-logistic model in addition to
GSSM) when back-calculating risk values. One can do so by taking a Bayesian modelling
approach and increasing the number of simulated scenarios, for example. However, care must be
taken as such transition may require an extensive computational time, careful consideration of
model convergence criteria, and challenges in creating simple yet informative summary figures.
We explicitly did not include the above in this study as our main objective was to create an
illustrative, yet convincing example to convey the importance of combining a simulation-
estimation exercise along with data fitting. The applicability of this approach is therefore not
only limited to GSSM but to all other statistical models where simulation-estimation exercise can
be performed“

The second criticism is about the decision to restrict the simulations to a single model type,
which is Gompertz State Space Models (GSSM). This wouldn't have been an issue if the goal
was to only explore the inherent biases with GSSM. However, the authors go on to provide a
“most plausible parameter value” depending on these simulations, and also generalize their
results for conservation management by calculating an index, again using these simulations. The
authors present no evidence that 627 population time series data is the result of a process similar
to GSSM. Concurrently, the bias structure might change if data simulations were done with a
different model than GSSM but it's analysis was still done with GSSM. The results reported,
then, is conditional on the fact that GSSM is a good approximation of the natural processes that
lead to all of the 627 population time series. This is a big assumption and is not mentioned in the
manuscript. Authors reference GSSM to be widely used in the ecological literature but none of
the cited papers actually compare GSSM to other model types that may include explicit
parameters for inter-specific competition, predation, or lag effects in density dependence, or that
may simply have a different structure (e.g. theta-logistic). I'm not suggesting to add other model

types to the current manuscript, although I think this a good area to explore and can be
 mentioned as a future goal. The presentation of the results, however, should change and it should
 be made abundantly clear that every result found with the 627 time series data comes with the
 assumption that GSSM is a good approximation of the natural processes that led to these time
 series. There could be many (or maybe only some) species among the 627 that GSSM is not a
 good fit. So patterns found with growth rate and density dependence strength (Figures 2 and 3)
 present only a piece of the evidence with a significant assumption. The more interesting and
 novel point is the applicability and generality of the presented framework and not GSSM and its
 accompanying patterns. The clear message of the manuscript for me is: If we have a species with
 a time series data, and if we have a simple model that we can show is a good candidate for
 natural processes that generates this time series data, then we can use simulations to incorporate
 uncertainty and to correct inherent model biases in decision making for species conservation.
 The assumptions of this framework and its applicability to other statistical models should be
 more clearly stated.

→ Thank you for this important comment: Yes, you are definitely right that the most interesting
 and novel point of this paper is the general applicability of the framework we presented here. We
 made this now clearer in the text.

The text now reads: “[...] our main objective was to create an illustrative, yet convincing
 example to convey the importance of combining a simulation-estimation exercise along with data
 fitting. The applicability of this approach is therefore not only limited to GSSM but to all other
 statistical models where simulation-estimation exercise can be performed“

 We also made clearer in the text that the results of this study on the pattern on risk level and
 species characteristics is dependent on the assumption of a GSSM. We have now written that:
 “Using the example of the GSSM, we find that the underlying risk of biased population status
 estimate varies depending on the species life history and length of the time series (Fig S3, S6-
 S7).” [...] “(i) we find that the estimated risk of large bias in population status could be
 substantial biased for data-limited species with high growth potential, high population variability,
 and weak density dependence, when using the GSSM to provide advice for management”

 We have also expanded in the discussion on the importance of considering other types of models
 (e.g. theta-logistics, Ricker) for future studies.

The text now reads: “Concomitantly, future studies could also consider other sources of
 uncertainty such as parameter estimation uncertainty or model types uncertainty (e.g. using a
 theta-logistic model in addition to GSSM) when back-calculating risk values. One can do so, e.g.,
 by taking a Bayesian modelling approach and increasing the number of simulated scenarios.
 However, care must be taken as such transition may require an extensive computational
 resources, careful consideration of model convergence criteria may be needed, and the Bayesian
 approach may be associated with challenges in creating simple yet informative summary figures.
 We explicitly did not include the above in this study as our main objective was to create an
 illustrative and convincing example to convey the importance of combining a simulation-
 estimation exercise along with data fitting. The applicability of this approach is therefore not

only limited to GSSM but to all other statistical models where simulation-estimation exercise can
 be performed“

Coming back to the reviewer point on cited reference that compares GSSM performance to other
 models, there is the paper from Anderson et al. 2017 Black-swan events in animal populations.
 PNAS (reference 19 in our manuscript) that have compared difference in parameter estimates
 between a GSSM and a Ricker-logistic model, a Gompertz model with no density dependence
 and a Gompertz model with autocorrelated residual. Their study also used the GPDD dataset.
 The authors concluded that “that parameter estimates were not systematically altered” when
 using these different models compared to the GSSM.

I also find Figures 2 and 3 hard to understand. The definition of risk zones and point estimates
 are clear. The arrows, however, are a bit confusing. The arrows point the most plausible true
 parameter estimate. Because this parameter estimate is now “true”, it should not point to another
 risk zone. If it is true, there is no risk of bias. It seems as if though, for some species bias
 correction can make things worse by moving the estimate to a higher risk zone. The risk zone of
 a point estimate and it’s bias correction with simulations are two different processes and belong
 to different figures.

→ Sorry for the confusing figure: We tried clarifying the caption and the corresponding text in
 the main text. Moreover, we do not think that these two processes (bias correction and risk zone)
 are independent: For each dataset, we are trying to figure out the parameter combinations that
 could have led to this dataset. Point estimates (from fitting the GSSM) are one solution but based
 on the simulation-estimation, we know that these can come from other combinations i.e. what we
 called the true parameter space or, in other terms, bias correction. But for each true parameter
 combination (if the data actually comes from this parameter combination instead), there is the
 associated risk of biased population status estimate when using GSSM, hence a new risk level.

Some additional suggestions:

- It might be a good idea to break the sentence starting at line 49 to multiple sentences. I read it
 several times to understand the definition of the index.

→ Thank you for the suggestion: We have now broken up the sentence.

- There should be iii) at line 316, not ii).

→ Corrected

- The rows in Figure S5 are same with the previous graphs. Why are they named differently?

→ Thank you for noticing: We have now corrected it.

REVIEWERS' COMMENTS:

Reviewer #1 (Remarks to the Author):

Thank you for the point by point response letter and the revised manuscript. I believe you have adequately responded to and incorporated (where appropriate) the comments from the three reviewers and so I conclude that the points raised in the previous round of reviews have been satisfactorily addressed.

Reviewer #2 (Remarks to the Author):

The manuscript is much clearer now and the authors have addressed my comments satisfactorily.

Reviewer #3 (Remarks to the Author):

I find the changes made to the manuscript satisfactory and recommend it's publication.

REVIEWERS' COMMENTS:

Reviewer #1 (Remarks to the Author):

Thank you for the point by point response letter and the revised manuscript. I
believe you have adequately responded to and incorporated (where appropriate)
the comments from the three reviewers and so I conclude that the points raised
in the previous round of reviews have been satisfactorily addressed.

→ Thank you

Reviewer #2 (Remarks to the Author):

The manuscript is much clearer now and the authors have addressed my
comments satisfactorily.

→ Thank you

Reviewer #3 (Remarks to the Author):

I find the changes made to the manuscript satisfactory and recommend it's
publication.

→ Thank you
